# Pre-Memorization Train Accuracy Reliably Predicts Generalization in LLM Reasoning

## Abstract

When large language models (LLMs) are finetuned on reasoning tasks, they can either reduce their training loss by developing problem-solving abilities, or by simply memorizing target traces in the training data. Our work aims to better understand how this learning process shapes a model's ability to generalize. We observe that, while LLMs often perfectly memorize most target solution traces by the end of training, their predictions at intermediate checkpoints can provide valuable insights into their behavior at test time. Concretely, we introduce the concept of *pre-memorization train accuracy*: the accuracy of model samples for training queries prior to exactly reproducing reasoning traces in the training data. We find that the average pre-memorization train accuracy of the model is strongly predictive of its test performance, with coefficients of determination around or exceeding 0.9 across various models (Llama3-8B, Gemma2-9B), datasets (GSM8k, MATH), and training setups. Beyond this aggregate statistic, we find that the pre-memorization train accuracy of individual examples can predict the model's sensitivity to input perturbations for those examples, allowing us to identify examples for which the model fails to learn robust solutions. A natural application of this insight is in data curation. We find that prioritizing the collection of examples with low pre-memorization accuracy leads to 1.5-2x data efficiency compared to i.i.d. data scaling, and outperforms other standard data curation techniques.

## 1 Introduction

Large language models (LLMs) have demonstrated remarkable problem-solving capabilities, yet the mechanisms by which they learn and generalize remain largely opaque. For instance, consider a set of LLMs, each derived from the same pretrained model and finetuned on the same reasoning dataset but with varying learning rates (Fig. 1). While several of these models reach near-perfect accuracy on training data, their test performances were vastly different. Conventional wisdom attributes this gap in generalization to *memorization* during training (Zhang et al., 2021; Bousquet & Elisseeff, 2000). However, during finetuning, modern LLMs often learn to perfectly replicate the entire training dataset while maintaining strong generalization, which suggests that memorization alone may not explain how LLMs generalize. This raises the question: *what factors in an LLM's finetuning process lead to differences in its generalization behavior?*

To investigate this question, we focus on mathematical reasoning tasks, where models are trained to generate both a final answer and intermediate reasoning steps. Although each problem has a single correct answer, the reasoning steps in a target solution trace represent just one of many valid ways to solve a problem. Therefore, a model that has memorized the training data is likely to replicate the exact reasoning steps from the solution trace for a problem in the training data, while a model with general problem-solving skills may produce the correct final answer but follow a different reasoning path. By analyzing model responses on training queries, focusing on both the accuracy of the final answer and the similarity of the response to the target solution trace, we can gain insights into whether the model is memorizing training data or developing problem-solving abilities.

Our investigation reveals that analyzing model samples for training prompts at different stages of training can provide insight into the model's generalization (test-time) behavior. While LLMs can often perfectly recall the entire finetuning dataset by the end of training, achieving perfect accuracy and exactly matching target solution traces, we observe distinct behaviors across training examples

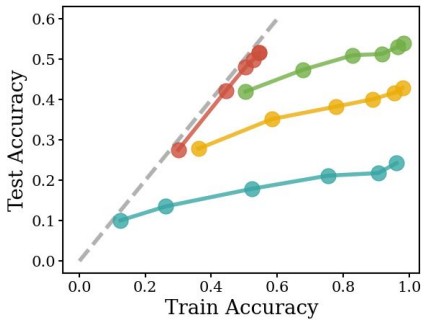 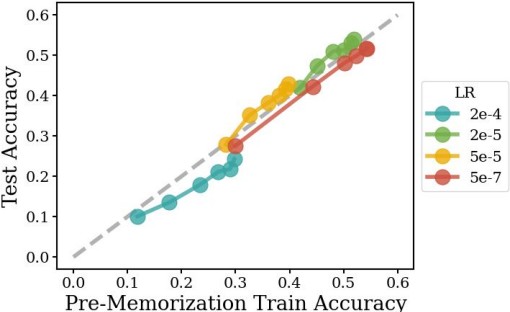

Figure 1: Relationship between train accuracy (left), pre-memorization train accuracy (right), and test accuracy for models finetuned on GSM8k using Llama3 8B. Each line represents a training run, and each point represents an intermediate checkpoint. Pre-memorization train accuracy strongly correlates with test accuracy, while train accuracy does not.

before full memorization occurs. For certain training examples, models only produce incorrect responses before memorizing the target response. For other examples, models first learn to generate diverse solution traces (distinct from the target solution trace) that all lead to the correct final answer, before later memorizing the target solution trace. Based on these observations, we introduce the concept of *pre-memorization train accuracy*, which refers to the highest accuracy a model achieves on a training example through the course of training before exactly memorizing the target solution trace. We find that a model's average pre-memorization train accuracy is highly predictive of the model's test accuracy, as illustrated in Fig. 1. Our experiments show that this phenomenon holds across different models (e.g., Llama3 8B (Dubey et al., 2024), Gemma2 9B (Team et al., 2024)), tasks (e.g., GSM8k (Cobbe et al., 2021), MATH (Hendrycks et al., 2021)), dataset sizes, and hyperparameter settings, with coefficients of determination around or exceeding 0.9.

Beyond predicting test accuracy, the pre-memorization accuracy of individual examples can provide insight into the robustness of model predictions for each example, which can guide training workflow decisions such as data curation. To investigate the robustness of a model's prediction, we present the model with training queries accompanied by a short preamble, phrases like "First" or "We know that", that deviate from the target solution trace. While the model often performs nearly perfectly on unaltered training prompts, its accuracy tends to drop considerably for certain examples when faced with these modified prompts. We find that examples with low pre-memorization train accuracy are much more likely to show reduced performance. This has natural applications in data curation: by selecting from data distributions that prioritize examples with low pre-memorization accuracy, we can increase the coverage of examples for which the model struggles to learn robust solution. Our experiments demonstrate that this approach leads to a 1.5-2x improvement in sample efficiency compared to i.i.d. data collection and outperforms other standard data curation techniques.

Our work provides a better understanding of the relationship between training dynamics and generalization in LLM reasoning. The main contributions of this work are as follows. First, we show that a model's pre-memorization train accuracy is highly predictive of test accuracy, providing a connection between a model's training process and its ability to generalize. Second, we show that pre-memorization accuracy can predict whether a model learns robust solutions for individual train examples, which offers practical implications for improving generalization such as in data curation.

## 2 RELATED WORKS

A number of works have studied the phenomenon of memorization during training, but consider different definitions of memorization. One definition quantifies memorization with "leave-one-out" performance, i.e., does the prediction of an example change significantly if we were to remove it from the training data? (Feldman & Zhang, 2020; Arpit et al., 2017; Zhang et al., 2021). While we do not use this definition due to its computational cost, the notion of pre-memorization accuracy in our work captures a similar concept, allowing us to identify examples that a model fails to robustly learn without conducting expensive leave-one-out evaluations. In the context of language models, others have defined memorized examples as those where the model's output closely matches examples in the training data (Carlini et al., 2021; Tirumala et al., 2022; Inan et al., 2021; Hans et al., 2024),

which has important privacy and copyright implications. Prior work has shown that this type of memorization is more likely to appear with duplicated data, larger model capacities, and longer context lengths (Carlini et al., 2022; Tirumala et al., 2022).

The relationship between the learning process and generalization has also been studied in a number of prior works. In particular, many works in this category focus on bounding the "generalization gap": the difference between training and test accuracies, using metrics related to model complexity, such as VC dimension or parameter norms (Neyshabur et al., 2015; Bartlett et al., 2019). Other works focus on empirically motivated measures, such as gradient noise (Jiang et al., 2019) or distance of trained weights from initialization (Nagarajan & Kolter, 2019), to predict generalization. Jiang et al. (2019) conducted a comprehensive comparison of these methods and found that none consistently predicted generalization, though their work primarily focused on image classification. Other approaches have used unlabeled, held-out data to predict generalization, leveraging metrics such as the entropy of model predictions or the disagreement between different training runs (Garg et al., 2022; Platanios et al., 2016; Jiang et al., 2021). Our findings suggest that pre-memorization accuracy can be a much stronger predictor of generalization in reasoning tasks with LLMs.

Finally, our work seeks to improve data curation, which has also been studied in a number of prior works. Active learning methods seek to optimally select data in an online learning setting for general machine learning models (Zhan et al., 2022; Gal et al., 2017; Tamkin et al., 2022). Specific to LLM finetuning, prior approaches largely fall into three categories: optimization-based, model-based, and heuristic-based approaches. Optimization-based methods frame data selection as an optimization problem, where the objective is model performance, and the search space consists of the training data distribution (Engstrom et al., 2024; Grosse et al., 2023). Model-based approaches, on the other hand, leverage characteristics of the learning process (Mekala et al., 2024; Liu et al., 2024), such as comparing the perplexity of examples (Li et al., 2023). Lastly, heuristic-based methods rely on simpler criteria, such as difficulty scores generated by GPT, to classify desirable training data (Chen et al., 2023; Lu et al., 2023; Zhao et al., 2023). Our data curation approach aligns most closely with model-based strategies, as we use the model's pre-memorization accuracy, a characteristic of the learning process, to inform the selection of training examples.

## 3 PRELIMINARIES

In this work, we focus on training LLMs to perform mathematical reasoning tasks via finetuning. We are provided with a training dataset $D_{\text{train}} = \{(x_i, y_i)\}$, where queries $x_i$ are drawn from $P(x)$ and solution traces $y_i$ are drawn from $P(y|x)$. We assume the test dataset, $D_{\text{test}}$, is generated similarly to the training data. The model is finetuned by minimizing next-token prediction loss. We denote the finetuned model as $f_\theta(y|x)$, and model predictions as $\hat{y} \sim f_\theta(y|x)$.

We further assume that solution traces consist of both intermediate reasoning steps and a final answer, denoted as $\text{Ans}(y)$. The goal of reasoning tasks is for the model to generate solution traces with the correct final answer when faced with previously unseen queries. We measure the accuracy of model samples for a given query $x_i$ using $\text{Acc}(f_\theta(y|x_i), y_i) = \mathbb{E}_{\hat{y}_i \sim f_\theta(y|x_i)}[\mathbb{1}(\text{Ans}(\hat{y}_i) = \text{Ans}(y_i))]$. In our experiments, we approximate this accuracy by sampling from the model with a temperature of 0.8 and averaging the correctness attained by the samples.

While different solution traces drawn from $P(y|x)$ should all have the same final answer, they may contain different reasoning steps. Thus the target solution trace $y_i$ of an example represents only one of many valid solution traces for solving $x_i$. We call an example *memorized* if the distance between the model's prediction and the target solution trace is low. Specifically, we consider $(x_i, y_i) \in D_{\text{train}}$ to be memorized by $f_\theta(y|x)$ if $\text{Perp}(f_\theta(y|x_i), y_i) < p$, where $\text{Perp}(f_\theta(y|x_i), y_i) = \exp(\frac{-1}{n_i} \log(f_\theta(y_i|x_i)))$, $n_i$ is the number of tokens in $y_i$, and $p$ is a threshold. We say the model has learned a *robust solution* to an example if, given $x_i$, the model is capable of producing solution traces different from $y_i$ which still arrive at the correct final answer.

## 4 CONNECTING THE LEARNING PROCESS TO GENERALIZATION

In this section, we investigate the relationship between a model's learning progression during finetuning and its capacity for generalization. Our findings reveal that, although models often memorize nearly the entire training dataset after some number of epochs, model-generated samples exhibit varying levels of accuracy over the course of training prior to memorization. Moreover, we find this

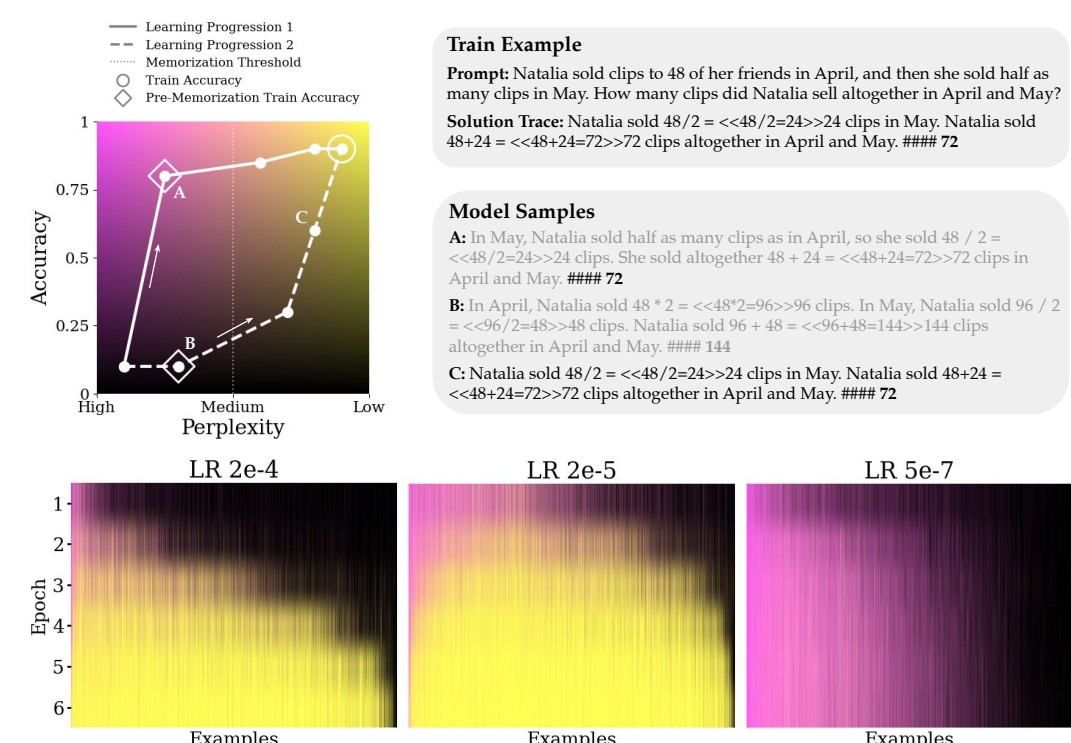

Figure 2: Visualizations of different learning progressions, as measured by the accuracy of model samples and the perplexity of target solution traces under model predictions. Top left presents a conceptual visualization, which represents accuracy with brightness and perplexity with color. Top right presents examples of model samples with (A) high accuracy+high perplexity, (B) low accuracy+high perplexity, and (C) high accuracy+low perplexity. Black text represents exact match with the target solution trace, while grey text represents parts that do not match. Bottom plots visualize the predictions of 3 different models through the course of training. The x-axis represents individual training examples, the y-axis represents the epoch of training, and the color represents model predictions for each example in terms of accuracy and perplexity (legend in top left visualization).

accuracy before memorization has a strong connection to the model's capacity for generalizing to new data. We discuss these findings in greater detail in the following sections.

## 4.1 CHARACTERIZING LEARNING PROGRESSIONS IN LLM REASONING FINETUNING

We begin by more precisely characterizing an LLM's learning process when finetuning on reasoning tasks. We focus on two key aspects of the model's behavior when presented with train queries: 1) whether the model's samples arrive at the correct final answer, and 2) the distance between the model's prediction and the target solution trace, measured by perplexity. These two metrics offer different perspectives on the model's behavior, because while there is only one correct final answer for each query, there may exist many different valid reasoning traces. Tracking both metrics through the course of training allows us to measure how effectively the model is able to solve training queries, and the extent to which this is accomplish by memorizing the target solution trace.

In Fig. 2, we visualize the learning progression, as characterized by the two metric described above, for three models finetuned on GSM8K. Each model is trained for three epochs, with a distinct peak learning rate that decays to zero by the end of training. As expected, training accuracy improves over time as the model minimizes the loss (color gradient from dark to light), and the distance between predictions and target solution traces decrease (from pink to yellow). For some learning rate settings, models approach near-perfect accuracy by the end of training, and their predictions closely match the target reasoning traces (mostly yellow in bottom row). However, during early stages of training, we observe significant differences in model behavior. For some examples, models initially produce incorrect predictions (black), and later replicate the target trace (yellow). For other examples, models first learn to generate correct answers with solution traces that differ from the target trace (pink), transitioning later to fully replicating the target trace (yellow).

We can see that with different training parameters, different models exhibit different capacities for generating accurate samples before memorizing target solution traces (amount of pink). For models with low accuracy before memorization, they may be largely learning verbatim mappings from training queries to target traces, which would not generalize to new queries. In contrast, models with high accuracy before memorization demonstrate an ability to arrive at correct answers through varied reasoning paths, suggesting that they have developed more generalizable problem-solving capabilities. We will more precisely study the connection between this behavior to test generalization in the next section, but first we introduce a new metric called pre-memorization accuracy to better quantify the accuracy of model samples before memorization.

We will use $f_{\theta_m}$ to denote the model at epoch $m$ of training, with $M$ as the total number of epochs. We first define a modified measure of accuracy as follows:

$$\text{Masked-Acc}(f_{\theta_m}(y|x_i), y_i, p) = \text{Acc}(f_{\theta_m}(y|x_i), y_i) \cdot \mathbb{1}[\text{Perp}(f_{\theta_m}(y|x_i), y_i) > p],$$

whose value is masked to zero if the model's prediction for that example is considered memorized (i.e., perplexity below $p$). We define the **pre-memorization accuracy** as follows:

$$\text{P-M Acc}(f_{\theta_{1:m}}(y|x_i), y_i, p) = \min\left\{ \max_{1 \leq m' \leq m} \text{Masked-Acc}(f_{\theta_{m'}}(y|x_i), y_i, p), \text{Acc}(f_{\theta_m}(y|x_i), y_i) \right\}$$

Unlike standard accuracy or masked accuracy, which evaluate performance at specific training checkpoints, pre-memorization accuracy evaluates the entire training process up to epoch $m$. We find that the average pre-memorization accuracy over the train data has a close relationship with the model's test accuracy. We will further elaborate on this relationship in the subsequent section.

## 4.2 Pre-Memorization Train Accuracy Strongly Predicts Test Accuracy

As discussed in the previous section, pre-memorization train accuracy reflects the quality of the model's predictions before it begins to memorize the training data. Similarly, test accuracy captures the model's performance on unseen test examples that are never memorized (by construction, since they are never trained on). If the training and test distributions of problems match, then one may expect a model's pre-memorization accuracy to roughly reflect its test accuracy, since both quantities are evaluated on data from the same distribution and neither capture inflated accuracies due to memorization. Indeed, our experiments confirm this intuition. We find that **a model's average pre-memorization train accuracy is highly predictive of its test accuracy** across a variety of training runs and checkpoints. More concretely, we find that there exists a value of $p$ for which a model's average pre-memorization train accuracy, $\mathbb{E}_{D_{\text{train}}}[\text{P-M Acc}(f_{\theta_{1:m}}(y|x_i), y_i, p)]$, closely approximates the model's test accuracy, $\mathbb{E}_{D_{\text{test}}}[\text{Acc}(f_{\theta_m}(y|x_i), y_i)]$. The value of $p$ is dependent on the task and pretrained model, but not dependent on training parameters.

We illustrate this relationship in Fig. 3, where we plot the pre-memorization training accuracy and test accuracy across different training runs. We used Llama3 8B and Gemma2 9B as base models and GSM8K and MATH as the reasoning tasks. To evaluate different generalization behaviors, we finetuned the models by adjusting the peak learning rate (ranging from 5e-7 to 5e-4), the number of training epochs (1, 3, 6), and the dataset size (full, half, or quarter of the original dataset). We use the same value for $p$ within each plot, and we find $p$ by sweeping across a range of values. A full list of the training runs in our experiments and their training details can be found in Appendix A. We observe a strong linear relationship between pre-memorization training accuracy and test accuracy, with the results closely following the $y = x$ line across different models, tasks, and hyperparameter settings. More quantitatively, the coefficients of determination associated with each plot are 0.94 (GSM8k Llama), 0.95 (MATH Llama), 0.97 (GSM8k Gemma), and 0.88 (MATH Gemma). Our results show that pre-memorization training accuracy is a reliable predictor of test accuracy.

As we discuss in Section 2, various metrics have been proposed in previous studies to predict the generalization gap, the difference between train and test accuracy. In Fig. 4, we compare several of these existing metrics, including gradient variance (Jiang et al., 2019), distance between current model weights and initialization (Nagarajan & Kolter, 2019), and an estimate of test accuracy via Average Thresholded Confidence (ATC) (Garg et al., 2022). We note that, while these approaches represent some of the most common approaches for predicting generalization in prior work, they do make use of different input assumptions compared to ours. We discuss our implementation of these metrics in Appendix B. The correlation coefficients associated with each metric (left to right)

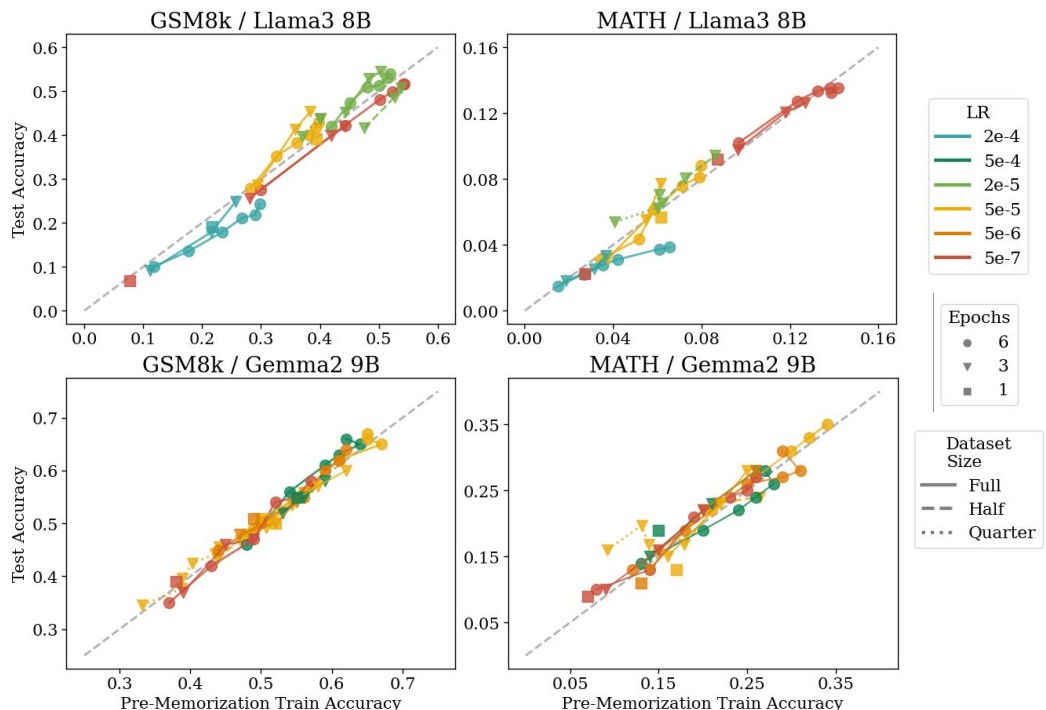

Figure 3: Evaluating the relationship between pre-memorization train accuracy and test accuracy. Each line corresponds to a training run, with each marker along the line representing a specific checkpoint. Pre-memorization train accuracy strongly predict test accuracy across tasks, models, and training settings.

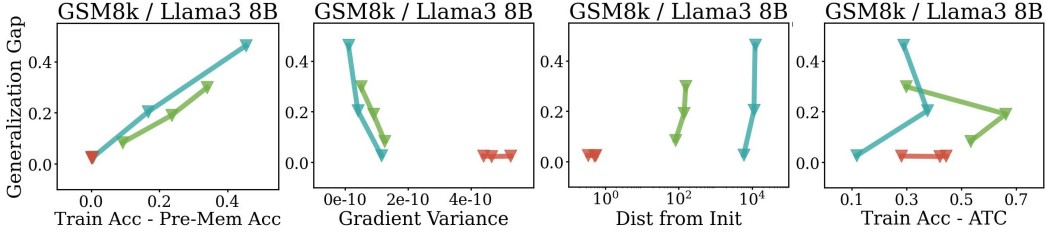

Figure 4: Evaluating different generalization metrics vs. the ground truth generalization gap for models finetuned on GSM8k using Llama3 8B (legend in Fig. 3). Our metric (left-most) is a much stronger predictor of the generalization gap than the other prior metrics.

are 0.98, -0.72, 0.59, -0.04, which shows that the prior metrics do not correlate as strongly with test accuracy as our proposed metric. A key advantage of our approach is that it leverages the assumption that model outputs include both reasoning steps and a final answer, enabling us to separate a model prediction's accuracy from its distance to the target solution trace. This distinction unveils the extent to which model's learn problem-solving solutions to training queries, which provides for a much more reliable estimate of a model's generalization abilities.

## 5 PER-EXAMPLE ANALYSIS OF GENERALIZATION

In the previous section, we demonstrated that a model's average pre-memorization accuracy is strongly correlated with its test accuracy, providing insight into the model's overall generalization capability. In this section, we go beyond aggregate test accuracy and show that tracking per-example

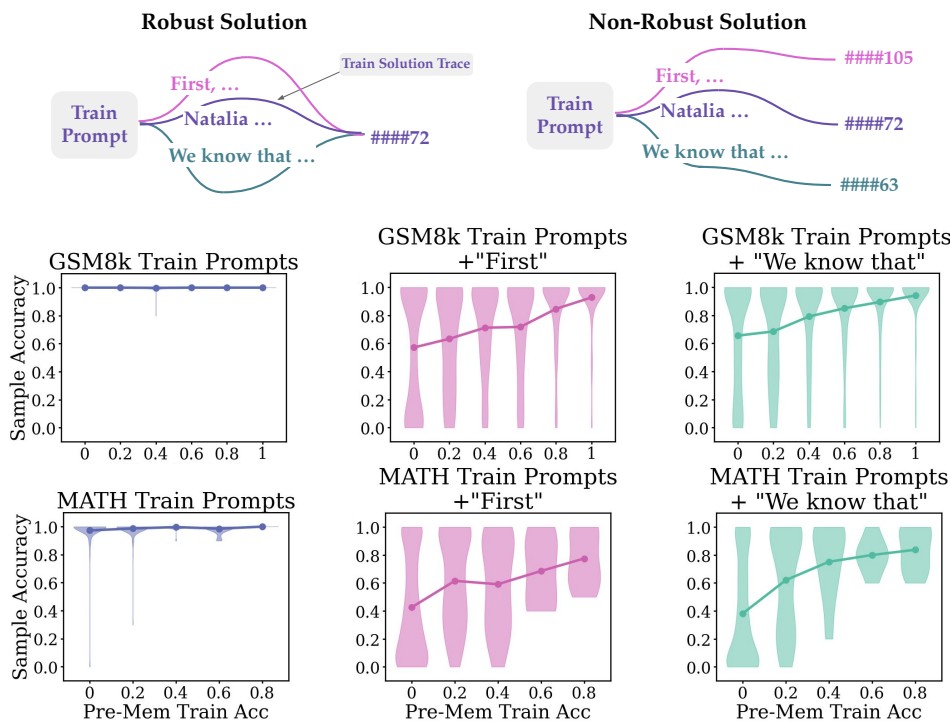

Figure 5: Visualization of the robustness of model predictions to perturbations in the prompt. Top left and right present conceptual visualizations of model responses when presented with the original training prompt (purple), original prompt + "First" (pink), and original prompt + "We know that" (teal). A robust model prediction would arrive at the correct final answer even if the perturbations changes the reasoning steps. In contrast, a non-robust model prediction produces incorrect final answer when the prompt diverges from the training data. Bottom plots present the accuracies of model samples (y-axis) when faced with the original prompt (left) and prompts with perturbations (middle, right). The x-axis represents the per-example pre-memorization train accuracy associated with each prompt. While the accuracy of model samples is almost perfect when faced with original prompts, it significantly degrades when faced with prompts with perturbations. Furthermore, the degradation of accuracy is much more significant for train examples with low pre-memorization accuracy than those with high pre-memorization accuracy, showing that per-example pre-memorization train accuracy can provide insight into the robustness of a model's individual predictions.

pre-memorization accuracy offers a window into the model's behavior at the level of individual training examples. Specifically, we find that the pre-memorization train accuracy of a given example is predictive of the model's sensitivity to input perturbations for that example. This example-level accuracy helps us identify subsets of the training data for which the model struggles to learn robust solutions and offers opportunities to improve training through targeted interventions. We explore how this insight can inform data curation strategies, showing that prioritizing examples with low pre-memorization train accuracy during data collection can lead to significant improvements over IID data collection and other common data curation methods.

## 5.1 PREDICTING MODEL ROBUSTNESS WITH PRE-MEMORIZATION TRAIN ACCURACY

We begin by examining the relationship between an individual example's pre-memorization train accuracy and the generalizability of the model's predictions for that example. Concretely, we assess how the solution learned by the model responds to small perturbations in the input prompt. We find that **model predictions tend to be less robust for train examples with low pre-memorization accuracy**.

---

**Algorithm 1** Our Data Collection Process

---

1: **Input:** $N' = N'_1 + \cdots + N'_n, t$
2: **Output:** Updated dataset $D'_{\text{train}}$
3: Initialize $D'_{\text{train}} = \{\}$
4: **for** $i = 1$ to $n$ **do**
5:     Train model on $D_{\text{train}} + D'_{\text{train}}$
6:     Evaluate model on $D_{\text{train}}$ and compute pre-memorization accuracy for each example
7:     Set $P'_i(x)$ as the distribution of examples with pre-memorization accuracy below $t$
8:     Collect $N'_i$ new examples from $P'_i(x)$ and add them to $D'_{\text{train}}$
9: **end for**

---

We visualize these findings in Fig. 5, which shows the behavior of two models, both trained for six epochs with a learning rate of 2e-5, on the GSM8K and MATH datasets. We present the model with both the original training queries, as well as training queries appended with short preambles to the solution trace—phrases such as "First" or "We know that"—which deviate from the target solution trace. We can see that while model performance is near-perfect for unaltered training prompts, certain examples exhibit a significant degradation in accuracy when presented with perturbed prompts. Furthermore, we can see that the accuracy of train examples with low pre-memorization train accuracy tend to degrate much more than those with high pre-memorization train accuracy. Additional experiments supporting this observation can be found in Appendix C.

Ideally, if the model has learned a robust problem-solving strategy, it should still be able to produce a valid reasoning trace, even with minor prompt deviations. Our findings suggest that pre-memorization train accuracy is a reliable indicator of the model's ability to generalize beyond memorization, and it can be used to identify fragile examples where the model may have learned overly specific or non-generalizable patterns. This insight into example-level fragility offers practical applications for improving model generalization.

## 5.2 CURATING DATA WITH PRE-MEMORIZATION TRAIN ACCURACY

We now present data curation as a practical application of the per-example understanding of generalization provided by pre-memorization train accuracy. While previous works have shown that prioritizing "hard" examples over "easy" ones during data collection can lead to more efficient scaling using heuristic measures of difficulty, the ideal metric for determining difficulty remains unclear. Our findings demonstrate that **pre-memorization accuracy can serve as a principled and more effective metric of example difficulty in data curation**.

**Problem setup.** We will first more precisely define our data curation problem. Given an existing set of $N$ training examples with queries distributed as $P(x)$, we aim to collect $N'$ examples, denoted as $D'_{\text{train}}$, to augment the dataset. The goal is to specify a new distribution $P'(x)$ that maximizes the test performance of a model trained on both the original and the newly collected examples. While defining the true distribution of queries can be challenging, we assume that by approximating it with an empirical distribution from the current dataset, we can collect new data with similar properties. In our experiments, we take $D_{\text{train}}$ to be the original dataset, and collect new examples by using GPT to rephrase examples in the original dataset, similar to the procedure in Setlur et al. (2024). By only collecting new examples that derive from the specified empirical distribution, we can ensure the new dataset approximates $P'(x)$. This setup can also be used when collecting new human-generated data, by providing the specified empirical distribution of examples as references for human labelers.

**Our approach.** We propose a data collection strategy that focuses on examples with low pre-memorization accuracy in the existing dataset. First, we calculate the pre-memorization accuracy for each example in the current dataset and then define $P'(x)$ as the distribution of examples whose pre-memorization accuracy falls below a certain threshold $t$. We then collect new data according to this distribution. If $N'$ is large, we can split the data collection process into multiple iterations ($N'_1 + ... + N'_n = N'$). In each iteration, we collect $N'_i$ new examples according to $P'_i(x)$, retrain a model on the combined dataset, calculate the pre-memorization accuracy with the model, and update $P'_{i+1}(x)$ for the next round of data collection. This process is summarized in Algorithm 1.

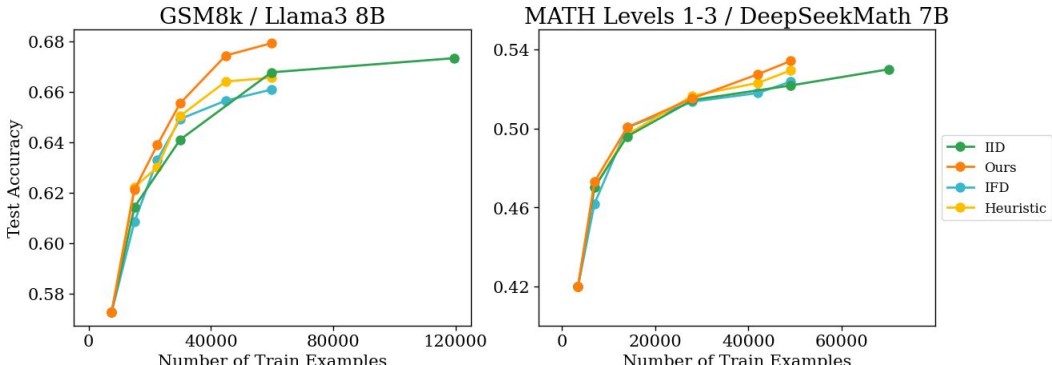

Figure 6: Comparison of different approaches for data curation. Each line represents a different data curation approach at varying scales of training dataset size, and each point represents a different training run. Our approach acheived the best sample efficiency compared to the other approaches.

**Comparisons.** We compare our strategy to IID sampling and two existing approaches commonly used in data curation. Both of these approaches propose a metric of example difficulty and prioritize difficult examples during data collection. The first metric, called Instruction-Following Difficulty (IFD) (Li et al., 2023), computes the ratio between the perplexity of training labels given inputs and the perplexity of only labels using a model finetuned for the task. The second metric uses external sources, such humans or more capable models such as GPT, to assign a heuristic notion of difficulty to each example (Chen et al., 2023; Lu et al., 2023; Zhao et al., 2023). For the GSM8K dataset, we use the number of lines in the target solution traces as a heuristic for difficulty, while for the MATH dataset, we use the difficulty levels provided in the dataset itself.

**Results.** In our experiments, we finetune Llama3 8B on GSM8K and DeepSeekMath 7B (Shao et al., 2024) on MATH levels 1-3 to evaluate the impact of our data collection approach on test accuracy. More details about the implementation of the different approaches can be found in Appendix D. As shown in Fig. 6, our approach outperforms all three prior approaches, achieving more than 2x the sample efficiency in reaching the same test accuracy compared to IID scaling in GSM8k, and more than 1.5x sample efficiency in MATH levels 1-3. These results highlight the effectiveness of using pre-memorization accuracy as a criterion for targeted data collection, leading to enhanced generalization with fewer data points.

## 6 CONCLUSION

Our work studies the relationship between training dynamics, memorization, and generalization in LLMs finetuned for reasoning tasks. We introduce the concept of pre-memorization train accuracy, the accuracy of model samples before they replicate target reasoning traces, and show that a model's average pre-memorization train accuracy is a strong predictor of its test accuracy. We further show that a model's per-example pre-memorization train accuracy can be an indicator of the robustness of a model's learned solution for those examples. We leverage this insight for data curation, and show that prioritizing examples with low pre-memorization train accuracy can be more effective than i.i.d. data scaling and other data curation techniques.

While our work demonstrated the strong connection between pre-memorization train accuracy and generalization, it is not yet clear how different training factors influence a model's pre-memorization train accuracy. Gaining a more precise grasp of this would allow us to better control and improve a model's generalization capabilities. Furthermore, our work focuses on reasoning problems, which admit a binary final-answer checking accuracy metric. It would be interesting to better understand the relationship between training dynamics and generalization in other more open-ended problem domains without a binary notion of correctness (e.g., creative writing, summarization, safety).

## 7 REPRODUCIBILITY STATEMENT

Our work uses standard supervised-finetuning techniques to train open-sourced models on well-known datasets. The main conceptual contribution of our work, the concept of pre-memorization train accuracy, is clearly defined in Sec. 4.1. Our Appendix includes additional details about the training hyperparameters used in our experiments and our implementations of prior approaches. We will release our code upon paper acceptance.

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

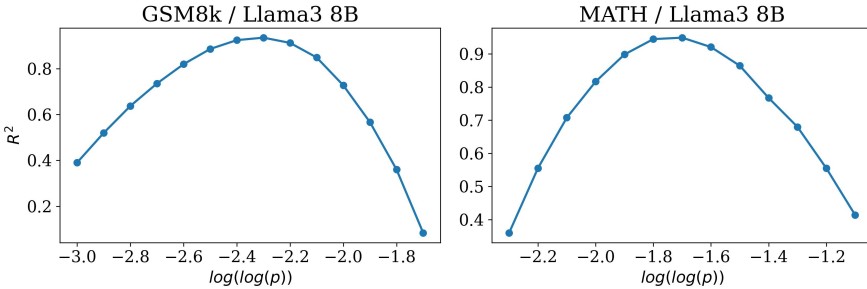

Figure 7: Relationship between the value of p and the coefficient of determination ($R^2$) with respect to pre-memorization train accuracy and test accuracy. The $R^2$ is taken in aggregate of all the corresponding training runs in Fig. 3.

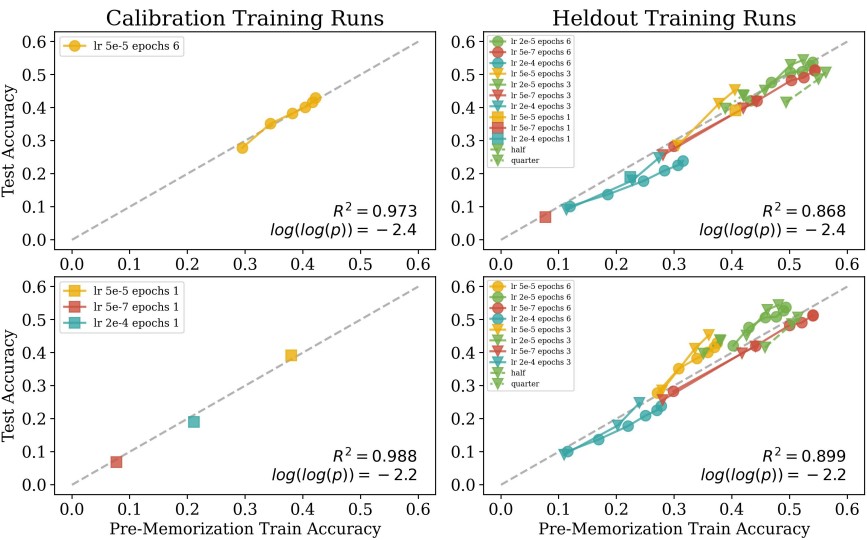

Figure 8: Calibrating $p$ on a subset of training runs, and evaluating $R^2$ on heldout training runs using GSM8k and Llama3 8B. We can see that calibrating on just 1-3 training runs was able to yield a robust value of $p$ which leads to high $R^2$ on heldout training runs.

## A  SECTION 4.2 TRAINING RUNS DETAILS

In this section, we will enumerate all training runs shown in Fig. 3 and their training details. For our half and quarter training runs, we fix the total number of training steps to be equivalent to training for 3 epochs on the full dataset.

### A.1  SELECTION OF MEMORIZATION THRESHOLD

We find the threshold $p$ by sweeping across a range of values, calculating the pre-memorization train accuracy across different training runs, and selecting the value which yields the strongest predictor of test accuracy. In Fig. 7, we illustrate how the value of $p$ influences the $R^2$ for predicting average test accuracy. We can see that $R^2$ degrades smoothly with respect to $p$, which makes it is relatively easy to find a good value of p by sweeping a range of values.

This calibration process only requires a small number of training runs (e.g. 1-3) to arrive at a robust value of $p$ which can generalize to new training runs on the same model and finetuning dataset, illustrated in Fig. 8. However, it is important the the training runs used for calibration exhibit some spread over test accuracies, and memorization during training.

Finally, we also show that the calibration process generalizes to new test examples. We divide the test set into two halves: a calibration test set, and a heldout test set. We calibrate $p$ on the calibration test set, and evaluate the coefficient of determination of the heldout test set. In Fig. 9, we can see

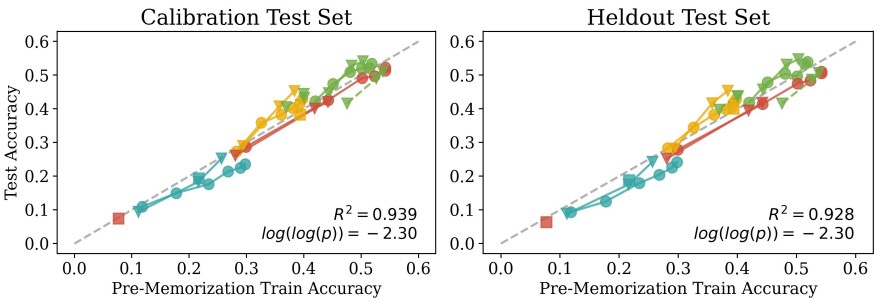

Figure 9: Calibrating $p$ using a subset of of the test set (calibration test set), and evaluating $R^2$ on a heldout test set using GSM8k and Llama3 8B. We can see that calibrating on just the calibration test set was able to yield a robust value of $p$ which leads to high $R^2$ on the heldout test set.

that the value of $p$ is able to generalize robustly to new examples on which it had not been calibrated, achieving high coefficient of determination.

## A.2   GSM8K LLAMA3 8B

For all training runs with GSM8k and Llama3 8B, we use the AdamW optimizer, with a linear decay learning rate scheduler with 20 warmup steps, a batch size of 128, and a max gradient norm of 2.

| Learning Rate | Epochs | Dataset Size |
|:---:|:---:|:---:|
| 5e-5 | 6 | full |
| 2e-5 | 6 | full |
| 5e-7 | 6 | full |
| 2e-4 | 6 | full |
| 5e-5 | 3 | full |
| 2e-5 | 3 | full |
| 5e-7 | 3 | full |
| 2e-4 | 3 | full |
| 5e-5 | 1 | full |
| 5e-7 | 1 | full |
| 2e-4 | 1 | full |
| 2e-5 | 6 | half |
| 2e-5 | 12 | quarter |

## A.3   MATH LLAMA3 8B

For all training runs with MATH and Llama3 8B, we use the AdamW optimizer, with a linear decay learning rate scheduler with 20 warmup steps, a batch size of 24, and a max gradient norm of 2.

| Learning Rate | Epochs | Dataset Size |
|:---:|:---:|:---:|
| 5e-5 | 6 | full |
| 5e-7 | 6 | full |
| 2e-4 | 6 | full |
| 5e-5 | 3 | full |
| 5e-7 | 3 | full |
| 2e-4 | 3 | full |
| 5e-5 | 1 | full |
| 5e-7 | 1 | full |
| 2e-4 | 1 | full |
| 2e-5 | 6 | half |
| 2e-5 | 12 | quarter |

## A.4 GSM8K GEMMA2 9B

For all training runs with GSM8k and Gemma2 9B, we use the Adam optimizer, with a cosine decay learning rate scheduler with (0.1*total steps) warmup steps, a batch size of 32, and a max gradient norm of 1.

| Learning Rate | Epochs | Dataset Size |
| --- | --- | --- |
| 5e-4 | 6 | full |
| 5e-5 | 6 | full |
| 5e-6 | 6 | full |
| 5e-7 | 6 | full |
| 5e-4 | 3 | full |
| 5e-5 | 3 | full |
| 5e-6 | 3 | full |
| 5e-7 | 3 | full |
| 5e-4 | 1 | full |
| 5e-5 | 1 | full |
| 5e-6 | 1 | full |
| 5e-7 | 1 | full |
| 5e-5 | 6 | half |
| 5e-5 | 12 | quarter |

## A.5 MATH GEMMA2 9B

For all training runs with MATH and Gemma2 9B, we use the Adam optimizer, with a cosine decay learning rate scheduler with (0.1*total steps) warmup steps, a batch size of 32, and a max gradient norm of 1.

| Learning Rate | Epochs | Dataset Size |
| --- | --- | --- |
| 5e-4 | 6 | full |
| 5e-5 | 6 | full |
| 5e-6 | 6 | full |
| 5e-7 | 6 | full |
| 5e-4 | 3 | full |
| 5e-5 | 3 | full |
| 5e-6 | 3 | full |
| 5e-7 | 3 | full |
| 5e-4 | 1 | full |
| 5e-5 | 1 | full |
| 5e-6 | 1 | full |
| 5e-7 | 1 | full |
| 5e-5 | 6 | half |
| 5e-5 | 12 | quarter |

# B SECTION 4.2 PRIOR GENERALIZATION METRICS

In this section we will more precisely describe each generalization metric.

## B.1 GRADIENT VARIANCE

We calculate the gradient of the model for 5 different minibatches, take the variance across the 5 samples for each element of each weight matrix, and take the average over each element of the model weights.

## B.2 DISTANCE FROM INITIALIZATION

We calculate the squared difference between each element of the model weights at initialization and after finetuning, and take the sun across all elements.

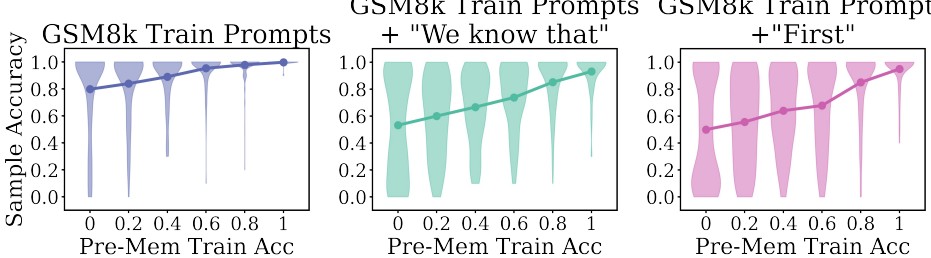

Figure 10: Accuracies of model samples (y-axis) when faced with the original prompt (left) and prompts with perturbations (middle, right). The x-axis represents the per-example pre-memorization train accuracy associated with each prompt. The accuracy of model samples tends to degrade more for those with low pre-memorization train accuracy when faced with perturbed prompts.

### B.3 AVERAGE THRESHOLDED CONFIDENCE (ATC)

ATC computes a threshold on a score computed on model confidence such that the fraction of examples above the threshold matches the test accuracy. For the score, we use the likelihood of greedily sampled responses under the model. We calculate the the score over the training data using a model trained for 3 epochs using learning rate 2e-5, and calculate the threshold over the score using the test dataset. We then predict the test accuracies over different models in our experiment by calculating the score associated with the training data using each model, and measuring the percentage of examples whose score surpass the threshold that we previously calculated.

## C SECTION 5 ADDITIONAL RESULTS

We reproduce our results from Fig. 5 for a Llama3 8B model trained on GSM8k with 3 epochs and a learning rate of 2e-5 in Fig 10. We can see that the accuracy of model samples tends to degrade more for those with low pre-memorization train accuracy when faced with perturbed prompts. Examples in the lowest bin (pre-memorization train accuracy $< 0.2$) showed average degradations of around $30\%$, whereas examples in the highest bin (pre-memorization train accuracy $> 0.8$) showed average degradations of less than $10\%$. This experiment shows that the results in Fig. 5 also hold of models trained for fewer numbers of epochs.

## D SECTION 5.2 IMPLEMENTATION DETAILS

For our approach for data curation, we implemented the process described in Algorithm 1, with 5 iterations ($n$) and using threshold ($t$) 0.75 for both GSM8k and MATH.

For the IFD approach for data curation, we calculated the IFD score using a model that was train on the test set associated each dataset for 2 epochs. This is because, in order to calculated the IFD score, we need a model which has been briefly trained for the task of interest, but which has not been exposed to the dataset for which we want to calculate the IFD score over. Note that this model is only used for calculating for the IFD score, and not used for evaluations in our experiments, so there is no data leakage.

For both the IFD approach and the heuristic approach, we take $P'(x)$ to be top 50 percentile of examples for GSM8k, and top 75 percentile of examples for MATH. We designed these percentiles to roughly match the percentile of examples that our approach selects from.

For all training runs, we use the AdamW optimizer, with a linear decay learning rate scheduler with 20 warmup steps, a batch size of 128, a max gradient norm of 2, a learning rate of 2e-5, and 3 epochs of training.

