# OpenReview forum: "Pre-Memorization Train Accuracy Reliably Predicts Generalization in LLM Reasoning"
_ICLR.cc/2025/Conference — Submitted to ICLR 2025_

### Official Review · Reviewer_JJKb · 2024-11-02

**Soundness:** 2
**Presentation:** 2
**Contribution:** 2
**Rating:** 3
**Confidence:** 4

**Summary:**

This paper focuses on investigating how the learning process influences the reasoning ability of LLMs to generalize. Specifically, this paper introduces the concept of pre-memorization train accuracy, which is shown to have a positive correlation with test accuracy. The authors also leverage this concept to develop a novel data curation method. Extensive experiments validate the effectiveness of the proposed model compared with a set of baselines.

**Strengths:**

The proposed method itself is coherent and easy to follow. The authors conduct experiments on several benchmarks to validate the effectiveness of the proposed method.

**Weaknesses:**

1.	Overall, the motivation behind the paper is not clearly articulated. The authors introduce a new concept, termed per-memorization train accuracy, which is calculated by sampling from the model multiple times and averaging the correctness of the samples. In my view, this metric is almost equivalent to the difficulty level of the problems. When the model samples multiple times and fails to obtain correct answers, it indicates that the problem is difficult. Conversely, successful sampling suggests the problem is easier. Furthermore, the difficult level of problems is often provided in many datasets, such as MATH. The authors should discuss this straightforward metric in the main text and include comparisons in the experiments.
2.	The paper provides insufficient explanation of why the proposed metric is expected to correlate positively with test accuracy. Additionally, it lacks theoretical justification to support the effectiveness of the proposed data curation method.
3.	The authors employ only two LLMs in their experiments. To strengthen the evaluation, recently proposed dense LLMs, such as Mistral and Qwen, as well as sparse Mixture-of-Experts (MOE) models like Mixtral and DeepSeekMOE, should be included as the backbone models for comparation.
4.	Reasoning is a general ability for complex problem-solving. Beyond mathematical reasoning, there are many other important reasoning tasks, such as logical reasoning, commonsense reasoning. To comprehensively evaluate the effectiveness of the proposed method, the authors should conduct experiments on logical reasoning tasks (such as LogiQA) and commonsense reasoning tasks (such as HellaSwag and Winogrande).

**Questions:**

1.	Could you involve a broader range of LLMs, encompassing both dense LLMs and sparse MOE models, to provide a more comprehensive demonstration of the proposed method’s effectiveness?
2.	Could you involve a wider variety of reasoning task to offer a more holistic demonstration of the proposed method’s effectiveness?

---

> ### Author Response · Authors · 2024-11-20
> **Response 1**
>
> We thank the reviewer for the feedback! In our rebuttal, we provide more detailed explanations of the motivation of our work, potential reasons behind our observations, and new results comparing our metric to sample accuracy. We hope that with these clarifications, we are able to better convey the value of our findings.
>
> > Overall, the motivation behind the paper is not clearly articulated.
>
> The main motivation of our work is to better understand the train/test performance gap in LLM reasoning tasks, in order to design more principled strategies for improving generalization. When finetuning LLMs, models frequently achieve perfect accuracy by the end of training, yet they often achieve vastly different test performances. This suggests that, beyond training optimization (the extent to which training loss can be reduced), the generalization behavior of a finetuned model is governed by the training dynamics (the manner in which the model changes through training). Thus, to more strategically improve generalization, we aim to first better understand how a model’s training dynamics shapes generalization.
>
> Our findings with pre-memorization train accuracy (a property of a model’s learning dynamics) allows us to bridge a model’s train and test behaviors, which can enable us to make more targeted training improvements. Our experiments with data curation shows that collecting data based on pre-memorization train accuracy can indeed lead to significant improvements to sample efficiency.
>
> We have updated our paper to more clearly articulate this motivation.
>
>
> > In my view, this metric is almost equivalent to the difficulty level of the problems. When the model samples multiple times and fails to obtain correct answers, it indicates that the problem is difficult.
>
> Indeed our metric can be viewed as a model-specific notion of difficulty. However, measuring the model’s accuracy alone does not provide a reliable metric of difficulty that correlates with downstream generalization performance. This is because models can produce correct answers to “hard” training examples by simply memorizing the target trace, without developing generalizable reasoning abilities. In contrast, our pre-memorization train accuracy metric is able to discriminate between purely memorized examples vs ones for which the model learns in a more robust manner.
>
> To further illustrate this point, let us consider using model accuracy to predict example difficulty at different points throughout training. At the end of training, models are often able to achieve nearly perfect accuracy on the training set, which does not allow us to discriminate between the difficulties of different examples. We can also use a model checkpoint in the middle of training to evaluate sample accuracy. To evaluate its effectiveness at predicting generalization, we compare the test accuracies of models at the end of training to the train accuracies after 1 epoch of training (https://imgur.com/a/UqQbzJe). We can see that approach achieves a R^2 around 0.5, whereas pre-memorization train accuracy has a R^2 of around or above 0.9. The accuracy at the first epoch is a worse predictor, because models both memorize and robustly learn examples at different points of training, so evaluating the accuracy at any particular epoch does not produce a reliable predictor of generalization/difficulty.

---

> > ### Author Response · Authors · 2024-11-20
> > **Response 2**
> >
> > > The difficult level of problems is often provided in many datasets, such as MATH. The authors should discuss this straightforward metric in the main text and include comparisons in the experiments.
> >
> > What is difficult for a model may not be the same as what is difficult for a human, as the difficulty of an example for a model is influenced by factors such as the pretraining and finetuning distributions, or even the model architecture. Thus, pre-memorization train accuracy serves as a more principled and effective metric for data curation.
> >
> > Our paper discuss heuristic notions of difficulty in both the related works section as well as the “comparisons” section of Section 5.2. Our scaling experiments in Fig. 6 compares again data curation using heuristic notions of difficulty, including the difficulty labels in the MATH dataset, and shows that our approach for data curation is more sample efficient.
> >
> > > The paper provides insufficient explanation of why the proposed metric is expected to correlate positively with test accuracy.
> >
> > When finetuned with different training parameters, different models exhibit different capacities for generating accurate samples before memorizing target solution traces (amount of  pink in Fig 2). For models with low accuracy before memorization, they may be largely learning verbatim mappings from training queries to target traces, which would not generalize to new queries. In contrast, models with high accuracy before memorization demonstrate an ability to arrive at correct answers through varied reasoning paths, suggesting that they have developed more generalizable problem-solving capabilities as tokens produced by these models can be very diverse while still preserving the semantic concepts needed to solve the problem. Thus, we would expect a model with high pre-memorization train accuracy to also have high test accuracy, which similarly captures a model’s capacity for general problem-solving. We have updated our paper to include this explanation.
> >
> > > The authors employ only two LLMs in their experiments. To strengthen the evaluation, recently proposed dense LLMs, such as Mistral and Qwen, as well as sparse Mixture-of-Experts (MOE) models like Mixtral and DeepSeekMOE, should be included as the backbone models for comparation.
> >
> > We unfortunately do not have the computational resources at the moment to run our experiments on another model. The models in our paper (Llama3 8B, Gemma 2 9B) are both dense models, and it would indeed be interesting to understand the behavior of sparse MOE models with respect to our observations. However, evaluating on two models has been widely accepted as sufficient for papers accepted for ICLR/ICML/NeurIPS.
> >
> > > Beyond mathematical reasoning, there are many other important reasoning tasks, such as logical reasoning, commonsense reasoning. To comprehensively evaluate the effectiveness of the proposed method, the authors should conduct experiments on logical reasoning tasks (such as LogiQA) and commonsense reasoning tasks (such as HellaSwag and Winogrande).
> >
> > The observations in our work are specific to tasks which involve 1) chain-of-thought reasoning steps, and 2) a unique final answer which is nontrivial to guess correctly (i.e. not multiple choice). This style of tasks is common in mathematical reasoning tasks. While benchmarks such as LogiQA, HellaSwag, and Winogrande can be broadly considered reasoning benchmarks, they do not exhibit the listed characteristics. We have updated our paper to be more explicit about the assumptions that we make on task structure.

---

> > > ### Comment · Reviewer_JJKb · 2024-11-24
> > >
> > > Thanks for the authors' response. I have no further questions. After reading the comments from other reviewers and considering the current version of the paper, I will maintain my evaluation score. However, I strongly the authors to conduct experiments on a broader range of LLMs and additional reasoning tasks. This would provide a more comprehensive assessment of the proposed model and help to strengthen the overall argument of the paper.

---

> > > > ### Author Response · Authors · 2024-11-25
> > > >
> > > > Thank you for the response.
> > > >
> > > > Do you have specific suggestions for benchmarks that align with the assumptions of our work?
> > > >
> > > > We would like to highlight that evaluating on two datasets is a common practice in LLM reasoning papers accepted at comparable venues. For example:
> > > >
> > > > - Quiet-STaR: Language Models Can Teach Themselves to Think Before Speaking (COLM 2024)
> > > > - RL on Incorrect Synthetic Data Scales the Efficiency of LLM Math Reasoning by Eight-Fold (NeurIPS 2024)
> > > >
> > > >
> > > > Aside from the number of models and datasets we evaluate (which we believe aligns with standard expectations for papers in this area), do you have any remaining concerns or feedback?

---

> > > > > ### Author Response · Authors · 2024-12-02
> > > > >
> > > > > Thank you for your engagement in the reviewing process so far! As the rebuttal process is coming to a close, we would appreciate if you could let us know if you have any further concerns and/or consider raising the score. Thanks again!

---

### Official Review · Reviewer_m3Sp · 2024-11-03

**Soundness:** 1
**Presentation:** 3
**Contribution:** 2
**Rating:** 3
**Confidence:** 4

**Summary:**

The paper seeks to understand generalisation in LLMs by introducing a metric that when applied to training examples correlate to test set performance for GSM8K and MATH. Moreover, this metric can used as proxy for example difficulty and in turn be used for curriculum learning and/or data curation.

**Strengths:**

The paper takes an interesting approach by using studying the CoT in GSM8K/MATH (traces as they call it) of a training example.
By focusing on the variance of traces while maintaining the correct answer, the authors have a very nice definition of overfitting.

**Weaknesses:**

There are not enough details on how `p` is chosen.

By inspecting Figure 1, one could argue that the procedure of "calibrating" a `p` is simply selecting a sort of projection onto the `x=y` line which then causes the calibration of pre-memorisation train accuracy to be correlated the test accuracy.

The different run curves w/ different learning rates are similar enough that calibrating `p` could simply cause the correlation to be high.

Moreover, finding such `p`, we have created a list of examples which correlate high to the test set. This is a form of leakage. Specially if it needs to be calibrate for each model and task.

**Questions:**

* Can you elaborate further on what the impact of `p` calibration above?
* What is the impact of varying `p` on the metric value and correlation. Perhaps, would we show that we can calibrate `p` on a hold-out set of the training-set and the experiments still hold

It would be great to understand the impact of this.

---

> ### Author Response · Authors · 2024-11-20
>
> We thank the reviewer for the feedback and questions! In our rebuttal, we provide an explanation of how we do the selection of p, and additional experiments demonstrating the robustness of our observations to p, as well as the generalizability of p to heldout examples and training runs. Please let us know if you have remaining questions or concerns.
>
> > There are not enough details on how p is chosen.
>
> We find the threshold p by sweeping across a range of values, calculating the pre-memorization train accuracy across different training runs, and selecting the value which yields the strongest predictor of test accuracy. We have added this explanation, as well as several experiments measuring the sensitivity of our observations to p, to Appendix A.1 in our submission.
>
> > What is the impact of varying p on the metric value and correlation.
>
> We conducted a sensitivity analysis of R^2 to values of p  (https://imgur.com/a/yGQrQ9C , Fig 7 in our paper), and found that the R^2 value exhibits a smooth degradation to the value of p, which makes it relatively easy to find a good value of p by sweeping a range of values.
>
> > Finding such p, we have created a list of examples which correlate high to the test set. This is a form of leakage…. Show that we can calibrate p on a hold-out set of the training-set and the experiments still hold.
>
> We present a new experiment splitting the test set of the original dataset into halves, calibrating p on one half of the test set (calibration test set), and evaluating the prediction results on the other half (holdout test set) (https://imgur.com/a/8X9WNEm , Fig. 9 in our submission). We can see that the value of p generalizes very well, achieving a R^2 of 0.928 on the holdout test set (only 0.01 below the calibration test set). Thus, this experiment shows that the high predictive power of pre-memorization train accuracy does not stem from any sort of data leakage from calibrating p.
>
>
> To further illustrate the robustness of p, we also experimented with splitting our training runs into calibration training runs and heldout training runs. We calibrate p only on the calibration training runs, and evaluate on the heldout training runs (https://imgur.com/a/HBDtzx3  , Fig. 8 in our paper). We can see that using only 1-3 training runs for calibration, we are able to arrive at a value of p which achieves a R^2 of >0.86 on heldout training runs. These experiments show that the high predictive power of pre-memorization train accuracy does not stem from “overfitting” p to specific training runs during calibration.

---

> > ### Author Response · Authors · 2024-11-25
> >
> > This is a gentle reminder that the discussion period is closing soon. In our rebuttal, we have included new experiments and clarifications which we believe address your main concerns. We would greatly appreciate it if you could take a look, and consider raising scores and/or engaging in further discussions. Thanks again!

---

> > > ### Comment · Reviewer_m3Sp · 2024-11-26
> > >
> > > I thank the authors for the reply.
> > >
> > > My main concern is that you have not demonstrated that there is no leakage of the test by fitting on a subset of the runs -- for example, the performance of these can be correlated. My splitting the test set in halves, you still don't rule correlation between those.
> > >
> > > Specially in such a narrow evaluation set -- I think it would need a much more comprehensive list of metrics to show
> > >
> > > By fitting a parameter to the test set, the burden of proof that there is no leakage is high. And there is such burden.
> > >
> > > I appreciate the reply but my score still stands.

---

> > > > ### Author Response · Authors · 2024-11-27
> > > >
> > > > Thanks for your reply. Would you be able to be more specific about the experiments you would like to see? We welcome suggestions for a “more comprehensive list of metrics to show”.
> > > >
> > > > We would further like to clarify that p is a single scalar value, akin to other hyperparameters like learning rate. It is standard practice to select hyperparameters on a validation set (calibrate test set in our case), and evaluate performance on a separate test set (heldout test set in our case). The performance on validation and test set will of course be correlated (they are generated from the same distribution), but there is no “leakage” as the two datasets are distinct. We believe the new experiments that we provided during the rebuttal period exactly matches the widely accepted protocol of using a validation set for selecting hyperparameters, and evaluating on a separate test set.

---

> > > > > ### Author Response · Authors · 2024-12-02
> > > > >
> > > > > Thank you for your engagement in the reviewing process so far! As the rebuttal process is coming to a close, we would appreciate if you could let us know if you have any further concerns and/or consider raising the score. Thanks again!

---

### Official Review · Reviewer_9AJC · 2024-11-04

**Soundness:** 2
**Presentation:** 2
**Contribution:** 2
**Rating:** 3
**Confidence:** 4

**Summary:**

This paper explores the problem-solving abilities and memorization processes of LLMs in the context of mathematical reasoning tasks. By introducing the concept of pre-memorization train accuracy, the authors establish a more effective metric for predicting the model's final test accuracy. This metric tends to serve as an indicator of training robustness and generalization capabilities. Additionally, it can be utilized to collect training data in a curriculum setting, resulting in a 1.5 to 2 times improvement in sample efficiency.

**Strengths:**

1. The paper highlights the discrepancy between training accuracy and test accuracy, as well as the impact of different training parameters (learning rate) on the training process, providing motivation for the methods proposed in the study.
2. The authors conduct experiments on Llama-3 8B and Gemma2 9B, aiming to demonstrate the generalizability of their method across different models.
3. The proposed method shows improvements in predicting test set accuracy and in curriculum learning scenarios compared to other baseline methods.

**Weaknesses:**

1. Many details are not adequately presented, such as the selection for the threshold $ p $ in pre-memorization train accuracy, as well as the contamination issues in the synthetic training data for Figure 6.
2. While the paper aims to investigate the model's generalization capabilities, all experiments utilize training and test sets from the same distribution, lacking out-of-distribution experiments.
3. In the experiments predicting testset accuracy, the comparisons with other methods are not entirely fair. The gradient variance and distance from initialization methods rely on significantly less input information during prediction, while the ATC method primarily focuses on OOD scenarios.

**Questions:**

1. In L246-247, the authors mention that the value of $ p $ is dependent on the task and the pretrained model, but I could not find any explanation about the selection of $ p $ in the paper. How is the value of $ p $ determined, and is the correlation coefficient for predicting testset accuracy sensitive to this parameter?
2. The authors claim that this training set accuracy metric based on the training process is superior, but they lack necessary ablation studies. For instance, how would the prediction performance change if only the training set accuracy after the first epoch(rather than first m epochs) is used?
3. The experiments in Figure 5 utilize 6 epochs of training, and it is not surprising to reach such conclusions in a setting that is sufficiently overfitted. In practice, it is uncommon to choose such a high number of epochs. Can the same conclusions be drawn with a maximum of 3 epochs?
4. In L413, it is mentioned that the data collection methods used in the paper can be applied to human data. Given that there are already some large-scale open-source datasets available, could the experiments be repeated on these datasets (e.g., Numina-CoT) to verify the reproducibility of the results?
5. The experiments in Figure 6 are conducted on GSM8K and MATH at difficulty levels 1-3. Considering that these datasets are relatively easy, I am curious about the experimental results at difficulty levels 4-5 in MATH.

---

> ### Author Response · Authors · 2024-11-20
> **Response 1**
>
> Thank you for your feedback! To address the concerns, we have provided an explanation of the selection of p, experiments showing the sensitivity of p, a comparison with train accuracy after 1 epoch, and a reproduction of Fig. 5 using a model trained with 3 epochs. Please let us know if this addresses the issues, or if there are further changes or clarifications that you would like.
>
>
> > Many details are not adequately presented, such as the selection for the threshold p  in pre-memorization train accuracy
>
> We find the threshold p by sweeping across a range of values, calculating the pre-memorization train accuracy across different training runs, and selecting the value which yields the strongest predictor of test accuracy. We have added this explanation, as well as several experiments measuring the sensitivity of our observations to p, to Appendix A.1 in our submission.
>
> > Is the correlation coefficient for predicting test set accuracy sensitive to this parameter?
>
> We conducted a sensitivity analysis of R^2 to values of p (https://imgur.com/a/yGQrQ9C , Fig 7 in our paper). We can see that the R^2 value exhibits a smooth degradation to the value of p, which makes it relatively easy to find a good value of p by sweeping a range of values.
>
>
>
> >The authors claim that this training set accuracy metric based on the training process is superior, but they lack necessary ablation studies.
>
> To the best of our knowledge, there are no prior works which have proposed generalization metrics with the exact same assumptions as ours. Instead, we compared our metric against a range of existing approaches which makes use of slightly different inputs (gradient variance, distance from initialization, and ATC). We have added a comparison to the accuracy at the first epoch (see below). Please let us know if there are any other ablations that you would like to see.
>
> > How would the prediction performance change if only the training set accuracy after the first epoch(rather than first m epochs) is used?
>
> We find the average train accuracy after the first epoch to predict the final test accuracy, with R^2 around 0.5 (https://imgur.com/a/UqQbzJe ).  This is because models often do not learn a subset of examples until later epochs. In contrast, pre-memorization train accuracy has a R^2 of around or above 0.9, making it a more reliable predictor of generalization.
>
> > In the experiments predicting test set accuracy, the comparisons with other methods are not entirely fair. The gradient variance and distance from initialization methods rely on significantly less input information during prediction, while the ATC method primarily focuses on OOD scenarios.
>
> We acknowledge that the comparisons with prior methods make use of different input information, though we do not believe they use significantly less information. While our approach makes use of model outputs (accuracy / perplexity) and requires calibration for the hyperparameter p, prior approaches like gradient variance and distance from initialization make use of information about the model weights, which ours does not. We also acknowledge that ATC was primarily designed for OOD settings, though we adapt their insights to the IID setting. We used these baselines because they are, to the best of our knowledge, the most well known/effective existing approaches for predicting generalization in prior works. We have added a note in our paper highlighting the differences in assumptions between our approach and these prior approaches.

---

> > ### Author Response · Authors · 2024-11-20
> > **Response 2**
> >
> > > The experiments in Figure 5 utilize 6 epochs of training, and it is not surprising to reach such conclusions in a setting that is sufficiently overfitted. In practice, it is uncommon to choose such a high number of epochs. Can the same conclusions be drawn with a maximum of 3 epochs?
> >
> > Yes. We conducted the same experiment in Fig. 5 using a model trained for 3 epochs using a learning rate of 2e-5 (https://imgur.com/a/6nzldbM , Fig 10 in our paper). We can see that the accuracy of model samples tends to degrade more for those with low pre-memorization train accuracy when faced with perturbed prompts. Examples in the lowest bin (pre-mem acc < 0.2) showed average degradations of around 30%, whereas examples in the highest bin (pre-mem acc > 0.8) showed average degradations of less than 10%. This experiment shows that the results in Fig. 5 also hold of models trained for fewer numbers of epochs.
> >
> >
> > > Given that there are already some large-scale open-source datasets available, could the experiments be repeated on these datasets (e.g., Numina-CoT) to verify the reproducibility of the results?
> >
> > Unfortunately, we do not have the computational resources to rerun our scaling experiments at the moment, as they require running many different training runs (~150 GPU hours on an A100 for each model/task). However, the scaling experiments in our paper does not make use of any assumption specific to model generated examples, so we expect our results to also be replicable to other sources of data.
> >
> > > The experiments in Figure 6 are conducted on GSM8K and MATH at difficulty levels 1-3. Considering that these datasets are relatively easy, I am curious about the experimental results at difficulty levels 4-5 in MATH.
> >
> > MATH is a relatively difficult dataset, and the accuracy for level 4-5 examples tends to be very low for models at the 8-9B parameter scale. This makes it difficult to distinguish between the efficacy of different data collection approaches for level 4-5 examples. We focus on levels 1-3 in order to highlight the differences between different data collection approaches.
> >
> >
> > > While the paper aims to investigate the model's generalization capabilities, all experiments utilize training and test sets from the same distribution, lacking out-of-distribution experiments.
> >
> > Our paper focuses on the train vs test generalization gap, rather than the in-distribution vs out-of-distribution generalization gap. We have updated the introduction of our paper to make this more clear.

---

> > > ### Author Response · Authors · 2024-11-25
> > >
> > > This is a gentle reminder that the discussion period is closing soon. In our rebuttal, we have included new experiments and clarifications which we believe address your main concerns. We would greatly appreciate it if you could take a look, and consider raising scores and/or engaging in further discussions. Thanks again!

---

> > > > ### Comment · Reviewer_9AJC · 2024-11-26
> > > >
> > > > Thanks for the authors' response.
> > > > * For parameter $p$ : Thank you for providing more details. This paper highlights "pre-memorization train accuracy" as a primary contribution and states that it can predict test set accuracy. However, it is quite weird that  $p$ is chosen by fitting $R^2$ with a series of experiments and then using the fitting performance as an evaluation metric. Additionally,  $p$ needs to be reselected under different data distributions. In practical applications, training data often requires multiple iterations. How should this metric guide experiments?
> > > > * For the use of multiple epochs in metric: In the curve of fitting performance changes with $p$ provided by the author, $R^2$ sometimes reaches quite low values (<0.4), making it difficult for me to judge when using average train accuracy only after the first epoch to predict the final test accuracy ($R^2$=0.5). Could the author also plot a similar curve of fitting performance changes with $p$ only after the first epoch to illustrate the importance of using multiple epochs?
> > > > * For difficulty levels 4-5 in MATH: Given that elementary and middle school level math problems have almost been resolved, if the method cannot be proven to extend to more difficult problems, it will result in a loss of soundness.

---

> ### Author Response · Authors · 2024-11-27
>
> We thank the reviewers for the response. To address these concerns, we present new experiments showing that the value of p generalizes to held out test sets and training runs, as well as a comparison to the first epoch training accuracy with p.
>
> > It is quite weird that p  is chosen by fitting R^2 with a series of experiments and then using the fitting performance as an evaluation metric.
>
> We present new experiments showing that p does not need to be fitted on the dataset on which we ultimately evaluate on. The value of p generalizes quite well to heldout test datasets, as well as heldout training runs.
>
> First, we split the test set of the original dataset into halves, calibrating p on one half of the test set (calibration test set), and evaluating the prediction results on the other half (holdout test set) (https://imgur.com/a/8X9WNEm , Fig. 9 in our submission). We can see that the value of p generalizes extremely well, achieving a R^2 of 0.928 between pre-memorization train acc and test acc evaluated on the holdout test set. Thus, this experiment shows that the high predictive power of pre-memorization train accuracy does not stem from any sort of data leakage from calibrating p.
>
> We also experimented with splitting our training runs into calibration training runs and heldout training runs. We calibrate p only on the calibration training runs, and evaluate on the heldout training runs (https://imgur.com/a/HBDtzx3  , Fig. 8 in our paper). We can see that using only 1-3 training runs for calibration, we are able to arrive at a value of p which achieves a R^2 of >0.86 on heldout training runs. These experiments show that the high predictive power of pre-memorization train accuracy does not stem from “overfitting” p to specific training runs during calibration.
>
> > p needs to be reselected under different data distributions. In practical applications, training data often requires multiple iterations. How should this metric guide experiments?
>
> In Section 5.2 of our work, we discuss how pre-memorization train accuracy can be used to guide data curations. Our experiments show that our approach leads to a 1.5-2x improvement in sample efficiency compared to IID data curation, and outperforms other standard data curation approaches.
>
> In these experiments, the data distribution changes as we scale up the amount of training data. However, we only calibrated the value of p once, using the smallest version of the dataset. If the task does not shift significantly, we find the value of p to generalize relatively robustly.
>
>
> > In the curve of fitting performance changes with p provided by the author, R^2 sometimes reaches quite low values (<0.4), making it difficult for me to judge when using average train accuracy only after the first epoch to predict the final test accuracy (R^2=0.5). Could the author also plot a similar curve of fitting performance changes with p only after the first epoch to illustrate the importance of using multiple epochs?
>
> We experimented with using the metric $\text{Acc}(f_{\theta 1}(y|x_i), y_i) * 1[\text{Perp}(f_{\theta 1}(y|x_i), y_i) > p]$ to predict test accuracy, where $f_{\theta 1}$ refers to the model after 1 epoch of finetuning (https://imgur.com/a/Yfm0vUe). Sweeping across different values of p, we find that the R^2 values does not surpass 0.51. The most optimal values of p is when $1[\text{Perp}(f_{\theta 1}(y|x_i), y_i) > p] = 1$ for all examples. Thus, pre-memorization train accuracy, with R^2 >= 0.9, serves as a much better predictor of generalization.
>
>
>
> > For difficulty levels 4-5 in MATH: Given that elementary and middle school level math problems have almost been resolved, if the method cannot be proven to extend to more difficult problems, it will result in a loss of soundness.
>
> The reason we do not perform our data curation experiments with levels 4-5 in MATH is because they are too difficult for the size of models that we use (8-9 billion parameters). This does not mean that our findings do not generalize to difficult problems in general. If we were able to train with bigger, more capable pretrained models, then they would be able to achieve higher performance on levels 4-5, which would provide better signal for our comparisons.
>
> For academic papers in the field of LLMs, it is standard practice for researchers to experiment on smaller models with simpler tasks (where there is more signal), due to computational constraints. It is generally accepted that experimental results from 8-9 billion parameter models are likely transfer to bigger models on harder tasks.

---

> > ### Comment · Reviewer_9AJC · 2024-11-27
> >
> > Thank you to the author for providing more experimental results. I still have some question:
> >
> > The premise of predicting model's generalization performance for other data distributions is that there is no labeled data to use from that distribution, which is also the setting for methods you compared, such as ATC. In your method, since the parameter $p$ needs to be fitted on the validation set, why not directly use the model's accuracy on this validation set to make predictions?  If we can already obtain the accuracy of the validation set, why we need to design a method to predict the test set accuracy through the training set accuracy?

---

> > > ### Author Response · Authors · 2024-11-28
> > >
> > > Thanks for your response!
> > >
> > > Pre-memorization train accuracy provides a more detailed and actionable measure of model generalization than validation accuracy. This is because validation accuracy is a single number which captures the overall generalization capabilities of a model, whereas pre-memorization train accuracy can be measured for *individual* training example, and provides insights into how robustly a model has learned each example (Sec. 5.1). Furthermore, while validation accuracy is measurable (with a validation set), it does not describe *how* to improve generalization. In contrast, because training data is something that we can change, we can use pre-memorization train accuracy to guide improvements to the training strategy. In Section 5.2, we used pre-memorization train accuracy to guide data curation, and showed that this leads to better sample efficiency.
> > >
> > > Beyond applications in training improvements, our observations provide a deeper understanding of how training dynamics shape generalization. In particular, while it is commonly observed that models with perfect train accuracies can exhibit very different generalization behavior, it is not well understood why this happens. Our analysis of learning dynamics revealed that models have different mechanisms (directly memorization vs generalizable learning) for reducing training loss that lead to different downstream generalization behaviors. Pre-memorization train accuracy allows us to identify the kind of mechanism that a model has learned for each example, providing a bridge between a model’s learning and generalization behaviors.

---

> > > > ### Author Response · Authors · 2024-12-02
> > > >
> > > > Thank you for your engagement in the reviewing process so far! As the rebuttal process is coming to a close, we would appreciate if you could let us know if you have any further concerns and/or consider raising the score. Thanks again!

---

### Official Review · Reviewer_L8UJ · 2024-11-05

**Soundness:** 4
**Presentation:** 3
**Contribution:** 3
**Rating:** 8
**Confidence:** 3

**Summary:**

This paper studies how memorization impacts model generalization.  More precisely, the authors study how the pre memorization accuracy -- the accuracy before which an example is memorized by the model -- is particularly predictive of test performance.  The authors preform a series of experiments to quantify this phenomena, and show that it persists across different model scales and datasets.

**Strengths:**

Overall the experiments are convincing and the analysis is thorough.  The comparison to active learning metrics is particularly nice, although perhaps a slightly more principled set of metrics could have been chosen (e.g., an influence function that is a dot product of loss gradients).

**Weaknesses:**

There are no obvious weakness in the paper.   Some experiments would have been nice to see:
* in particular, having a more diverse set of model sizes would have been interesting.  Is the proposed metric still predictive for smaller models?  It seems that for small models, they may never end up memorizing the reasoning trace, and the notion of a pre-memorization accuracy does not exist.  Are there nevertheless proxy metrics available in those cases?  At what scale does the generalization emerge?
* It would be interesting to consider a more diverse set of active learning metrics.  For example, can influence functions be used to map examples to train data (as proposed in [1)?

[1] https://arxiv.org/abs/2308.03296

**Questions:**

(see above)

---

> ### Author Response · Authors · 2024-11-20
>
> We thank the reviewer for the nice feedback! We answer the question below.
>
>
> > It seems that for small models, they may never end up memorizing the reasoning trace, and the notion of a pre-memorization accuracy does not exist. Are there nevertheless proxy metrics available in those cases? At what scale does the generalization emerge?
>
> We find pre-memorization train accuracy to be predictive of test accuracy, regardless of whether the model memorizes the training data. One example is models trained with a learning rate of 5e-7, which, as illustrated in Fig. 2, does not memorize any training data by the end of training (no yellow in last row). In Fig. 3, we can see that the pre-memorization train accuracy of models trained with 5e-7 learning rate are still highly predictive of test accuracy. For situations where the model does not memorize the training data, the pre-memorization training accuracy is generally equivalent to the training accuracy.
>
>
> > It would be interesting to consider a more diverse set of active learning metrics. For example, can influence functions be used to map examples to train data (as proposed in [1])?
>
> We believe that influence function based approaches for data curation would likely be effective, but we did not implement this approach due to its computation and algorithmic complexity, such as the need to compute the inverse-Hessian-vector product for each query. In contrast, our approach only makes use of the model’s outputs, which allows us to treat the model as a “black box”. We believe simplicity is one of the advantages of our approach, which enables practitioners to quickly implement in their own training workflows.

---

### Author Response · Authors · 2024-12-04
**Rebuttal Summary**

Based on the rebuttal discussion, we have made the following improvements to our work:
1. We have provided a more in-depth explanation and analysis regarding the selection of the memorization threshold.
2. We have offered a clearer articulation of the motivation behind our findings and their implications.

After addressing the concerns raised and incorporating the suggestions, we believe we have resolved all the issues highlighted during the review. We are grateful to the reviewers for their constructive feedback, which has enhanced the rigor and clarity of our work.

---

### Meta-Review · Area_Chair_NsSJ · 2024-12-18

**Metareview:**

This paper studies the mechanisms by which LLM learn to solve reasoning tasks. In particular, they introduce the concept of "pre-memorization train accuracy" and use it to analyze the dynamics of model behavior during the course of training. They find that this notion is highly predictive of test performance and sensitivity to input perturbations, which they show can be used for effective subset data selection.

The main strengths of the paper are its introduction of the notion of pre-memorization accuracy and its use for test accuracy prediction. In turn, the main weaknesses are: limited clarity and unclear reproducibility, unclear motivation, lack of out-of-distribution experiments, and understated/underdiscussed hyperparameter choices.

Overall, the paper could use substantial improvement along several axes to solidify its contribution. Most importantly, I agree with reviewer JJKb that the motivation needs better articulation, and that the experimental validation should be much more extensive to prove the usefulness of the proposed metric. Hence, I side with the majority opinion of the reviewers in recommending rejection at this time.

**Additional Comments On Reviewer Discussion:**

The reviews for this paper prompted a lively discussion between the authors and (most of) the reviewers. The main points of that discussion were:

* Parameter ( p ) Calibration: Reviewers raised concerns about the selection and sensitivity of the threshold parameter ( p ), and whether its selection amounted to test-data leakage. In response, the authors introduced a modified experimental protocol that now split the test dataset into two halves and used only one of them to calibrate p. Additionally, they provided additional experiments showing the robustness of ( p ) and its generalizability to held-out datasets and training runs. But reviewer m3Sp countered that the modified protocol still relies on part of the original test dataset, which is likely highly correlated to the other half, and therefore does not truly demonstrate that there is not data leakage.
* Model and Task Diversity: Reviewers suggested evaluating the method on a broader range of models and reasoning tasks. The authors acknowledged this but cited computational constraints and highlighted that their findings are likely to generalize to larger models and more complex tasks.
* Theoretical Justification: Reviewers noted the lack of theoretical explanation for the observed correlations. The authors provided a more detailed discussion of the potential reasons behind their findings.
* Experimental Details: Reviewers requested more details on certain experimental setups, such as the selection of ( p ) and the impact of different training parameters. The authors added these details to the revised manuscript.

Among these, the issues of calibration and potential leakage of the p parameter and the narrow scope of evaluation were the main decisive factors in my final decision.

---

### Decision · Program_Chairs · 2025-01-22

Reject